# Three Preceding Crops Increased the Yield of and Inhibited Clubroot Disease in Continuously Monocropped Chinese Cabbage by Regulating the Soil Properties and Rhizosphere Microbial Community

**DOI:** 10.3390/microorganisms10040799

**Published:** 2022-04-10

**Authors:** Yiping Zhang, Wei Li, Peng Lu, Tianyu Xu, Kai Pan

**Affiliations:** 1Key Laboratory of Biology and Genetic Improvement of Horticultural Crops (Northeast Region), Ministry of Agriculture and Rural Affairs, Northeast Agricultural University, Harbin 150030, China; b210401014@neau.edu.cn (Y.Z.); liwei4562022@163.com (W.L.); lupeng18846054105@163.com (P.L.); xu15776323236@163.com (T.X.); 2College of Horticulture and Landscape Architecture, Northeast Agricultural University, Harbin 150030, China

**Keywords:** Chinese cabbage, previous crops, clubroot, soil microorganisms, rotation

## Abstract

Crop rotation can improve soil properties and is one of the important measures to prevent soil-borne diseases. This study aimed to evaluate the effects of different preceding crops on clubroot disease in Chinese cabbage and soil microorganisms, to provide a theoretical basis for the ecological control of clubroot scientifically. In this experiment, soybeans, potato onions, and wheat were used as the preceding crops and compared with the local preceding crop garlic. The growth of the Chinese cabbage, disease occurrence, soil chemical properties and changes in microbial community structure were determined by using quantitative real-time polymerase chain reaction (PCR), soil microbial high-throughput sequencing and other methods. The results showed that the rotation of potato onion and wheat with Chinese cabbage could reduce the clubroot disease index of Chinese cabbage remarkably. Through Illumina Miseq sequencing, when three previous crops were harvested, the abundance and diversity of the bacteria increased obviously, while the fungi decreased. The relative abundance of the phylum Proteobacteria and Firmicutes was strikingly reduced, while that of Chloroflexi was significantly increased. These results show that three previous crops changed the structure of soil microorganisms, reduced the clubroot disease of Chinese cabbage, promoted growth, and suppressed disease. The ranked effect on promoting growth and inhibiting diseases was potato onion > wheat > soybean.

## 1. Introduction

Clubroot disease is a serious soil-borne disease caused by *Plasmodiophora brassicae*, threatening the production of cruciferous plants worldwide [1]. In China, about 3.2–4.0 hectares of cruciferous crops are affected by clubroot per year, leading to yield reductions even crop failures [2]. In the prevention and control of diseases, ecological methods are particularly important for the prevention and treatment of clubroot.

Planting previous crops is one of the cultivation techniques used to alleviate the damage to crops by soil pathogens, which can reduce the chance of soil-borne pathogens invading roots, creating unfavorable environments [3]. Different crops have different effects on the inhibition of different soil-borne diseases through mechanisms such as interrupting the cycle of pathogens, allelochemicals, antagonizing microorganisms, and increasing soil carbon content by root exudates and residues to increase soil microbial biomass and activity [4].The rotation of legumes and gramineous crops with cruciferous crops can alleviate the occurrence of cruciferous clubroot [5], the application of rice straw is able to stabilize soil bacterial community composition and inhibit clubroot disease [6], and wheat root exudates increase the defense response and resistance of watermelon [7], as well as inhibiting the occurrence of watermelon wilt [8].

The physical and chemical properties of soil affect the growth of crops directly or indirectly. The long-term monoculture of crops results in the selective absorption of nutrients, which eventually leads to soil nutrient imbalance [9]. In cucumber continuous cropping systems, salt accumulates and soil properties deteriorate [10], soil pH first decreases and then increases, the content of soil nitrogen and phosphorus tends to increase, and potassium decreases [11]. The soil type, bulk density, water content, pH, organic matter content, ion concentration of calcium, silicon, and boron all affect the propagation of resting-spores or the occurrence of clubroot [12]. Long-term continuous cropping could be accompanied by changes in soil microorganisms (involvion an increase of harmful microorganisms), the loss of soil biomass, and an increase in crop diseases [13]. Peas and soybeans are continuously single-planted, resulting in changes in the soil microbial community, which affects the growth of plants [14]. Mustard and wild rocket are used as green fertilizers to improve cucumber fusarium wilt resistance by changing the composition of the cucumber rhizosphere bacterial community [15]. As a protozoan that infects cruciferous plants obligately [16], *P. brassicae* changes the microbial community’s structure in the rhizosphere soil when it infects cruciferous crops. Compared with the rhizosphere soil of normal plants, the abundance of Bacteroides in the rhizosphere soil of susceptible plants is significantly increased, while Nematomycota and Ascomycota are remarkably reduced. The relative abundance of Flavobacterium rhizosphere, Streptomyces, actinomycetes, and phototrophic bacteria decrease, while Bacteroides, fungal chytrid, Mortierella, and Basidiomycetes increase. When rhizobia infect cruciferous plants, they are also susceptible to infection by the chytrid phylum, which reflects the symbiotic relationship between the two microorganisms in the rhizosphere [17].

This experiment researched the effects of different previous crops and Chinese cabbage rotation on Chinese cabbage growth, disease occurrence, soil chemical properties and microorganisms. We speculate that crop rotation can improve soil environment, promote plant growth, and alleviate the effect of clubroot on Chinese cabbage.

## 2. Materials and Methods

### 2.1. Soil and Plant Preparation

An experiment was conducted in a field located in Baicheng Village, Acheng District, Harbin, China (45°41′ N, 126°45′ E) during 2018–2019. In this location, the farming pattern of planting garlic in spring and Chinese cabbage in autumn has been repeated for more than 30 years, and the incidence of clubroot has been severe in the past 10 years. The soil was sandy loam and contained pH 6.43, the value of electrical conductivity (EC) was 0.398 mS/cm, soil organic matter content (SOM) was 3.07%, alkali hydrolyzable nitrogen (AN) was 130.41 mg·kg^−1^, available phosphorus (AP) was 125.25 mg·kg^−1^, available potassium (AK) was 622.50 mg·kg^−1^, and total nitrogen (TN) was 2.51%. The content of resting spores of *P. brassicae* in soil was 1.95 × 10^5^ per gram.

The variety of Chinese cabbage (*Brassica rapa* subsp. *pekinensis*) was “Gai liang Dong bai No. 1”, potato onion (*Allium cepa* var. *aggregatum* Don.) was M29, wheat (*Triticum aestivum* L.) variety was “Dongnong 123”, soybean (*Glycine max* (Linn.) Merr.) variety was B101, and garlic (*Allium sativum* L.) was the local purple garlic of Acheng.

### 2.2. Experimental Design

The experiment was conducted from April 2018 to October 2019 and repeated for two consecutive years. There were four treatments in the experiment, including soybean rotation, potato onion, wheat, and garlic rotation as control: (1) soybean Chinese cabbage (B); (2) potato onion Chinese cabbage (O); (3) wheat Chinese cabbage (W); (4) garlic Chinese cabbage (CK). The experiment was randomly designed with three replicates per treatment, and each treatment plot was five ridges (3 m × 7 m). Potassium sulfate compound fertilizer (15-15-15, 10 kg·hm^−2^) and diammonium hydrogen phosphate (10 kg·ha^−2^) were applied as base fertilizer before sowing. Potato onions, wheat, and garlic were planted from 5–10 April, and soybeans were planted on about 28 April. After the previous crop was harvested, Chinese cabbage was sowed on about 23 July, with a row spacing of 50 cm × 50 cm, and harvested on about 5 October.

### 2.3. Determination of Growth and Disease

When the Chinese cabbage was planted for 40 days, 9 Chinese cabbage plants were randomly selected from each plot, the biomass of dry above-ground parts was measured, and the rate and incidence of clubroot were investigated. Nine Chinese cabbage plants were randomly selected from each plot after heading to determine the yield, which was converted into a per-plot value. The clubroot disease classification, incidence and disease index calculations of Chinese cabbage were based on Sharma Kalpana et al. [18].

### 2.4. Soil Collection

Soil was collected from the upper soil layer (0–15 cm) and sieved (2 mm) for DNA extraction and high-throughput sequencing after the preceding crops were harvested. To determine the concentration of the resting-spores of *P. brassica* and the changes in microorganisms when the previous crops were harvested, the physical and chemical properties were determined after drying naturally.

### 2.5. Determination of Soil Chemical Properties

The chemical properties of soil included soil pH, EC, organic matter, alkali hydrolyzed nitrogen, available phosphorus and available potassium content. The analysis of all indexes was based on Liu et al. [19].

### 2.6. DNA Extraction and Quantification PCR

All soil samples were stored at −80 °C. DNA was extracted from 250 mg of soil samples using a HiPure Soil DNA Kit according to the manufacturer’s instructions.

Quantification of 16S rRNA and ITS genes was performed on an iQ5 Real-Time PCR Detection System (Bio-Rad Lab, Hercules, CA, USA). The primer of the bacterial communities was 341F (CCTACGGGNGGCWGCAG)/806R (GGACTACHVGGGTATCTAAT); the primer of the fungal communities was ITS3_KYO2F (GATGAAGAACGYAGYRAA)/ITS4-2409R (TCCTCCGCTTATTGATATG).

For *P. brassicae*, the primer was based on Wallenhammar et al. [20]. The forward primer was AAACAACGAGTCAGCTTGAATGC, and the reverse primer was TTCGCGCACAAGCACTTG. Each 20 µL of PCR reaction contained 9 µL of 2 × Real SYBR mixture, 0.2 µL of each primer (10 µM), 2.5 µL of template DNA, and 8.1 µL of ddH2O. The qPCR reaction conditions were as follows: 5 min at 95 °C for initial denaturation, 22 amplification cycles of 50 s at 95 °C for denaturation, 30 s at 65 °C for annealing, 1 min at 72 °C for an extension, 10 min at 72 °C for a final extension, and an amplified fragment length of about 230 bp.

The amplified products were purified with AMPure XP beads, quantified with the ABI StepOnePlus Real Time PCR system (Life Technologies, made in Carlsbad, CA, USA), and sequenced according to the PE250 mode pooling of Hiseq2500.

### 2.7. Analysis of Soil Microbial Diversity

The original data were analyzed by Turkey’s HSD in SPSS 23.0 software at a *p* < 0.05 level. Pearson correlation analysis was also performed by SPSS 23.0. The bar diagram was prepared using Prism 8.00 software. The sequenced data were analyzed by Quantitative Insights Into Microbial Ecology (QIIME), Version 1.9.0. The community structures of bacteria and fungi were visualized by principal co-ordinates analysis (PCoA) with Bray-Curtis distance dissimilarity matrices to clarify the differences in the community compositions of bacteria and fungi in the different treatments. A beta diversity, RDA Mantel test was performed using the “vegan” package in the R environment (R Studio).

## 3. Results

### 3.1. Effects of Different Preceding Crops on Growth of Chinese Cabbage

The yield of the Chinese cabbage in the three treatments increased by more than 20% compared with the CK. The yield and the dry weight on the ground of the Chinese cabbage in the treatments of potato onion Chinese cabbage and wheat Chinese cabbage were significantly higher than the CK over the two years. In addition, the dry weight on the ground of the Chinese cabbage in 2019 was generally lower than in 2018 (*p* < 0.05) (Figure 1).

### 3.2. Effects of Different Preceding Crops on Incidence of Chinese Cabbage Clubroot Disease

In the two consecutive years, the disease index of clubroot disease in the three treatments was significantly lower than that of the control 40 days after the sowing of the Chinese cabbage; potato onion rotation had the lowest disease index (*p* < 0.05) (Figure 2).

### 3.3. Effects of Different Preceding Crops on Soil Chemical Properties

The preceding crops significantly increased the pH of the soil and reduced the EC compared with the control (*p* < 0.05). The wheat rotation increased the soil total nitrogen content and reduced the available potassium content obviously compared with the control and other treatments in 2018 (*p* < 0.05). In 2019, the preceding crops all reduced the available potassium content (*p* < 0.05); however, none of them affected the soil alkali hydrolyzable nitrogen or available phosphorus content, nor did they notably reduce the soil organic matter content (*p* > 0.05) (Table 1).

### 3.4. Effects of Different Preceding Crops on Resting Spores of P. brassicae

Sixty days after the preceding crop sowing, the *P. brassicae* resting spore content was significantly lower than the control in 2018 and 2019. (*p* < 0.05) (Figure 3).

### 3.5. Alpha and Beta Diversities of Bacterial and Fungal Communities in Soil

In this study, twelve samples of bacterial 16SRNA V3–V4 regions and twelve fungal ITS regions were subjected to Miseq sequencing. After flattening according to the minimum number of samples, a total of 2,209,500 sequences was obtained from the bacteria and 1,360,488 from the fungi.

The observable bacterial OTU number and Shannon index of the potato onion rotation treatment were significantly higher than the control and the other treatments after the preceding crops were harvested (*p* < 0.05), and the bacterial Shannon index of the three treatments was obviously higher than the control, indicating that the three treatments, especially the potato onion, could increase the number and diversity of bacterial communities compared with the control during planting (Figure 4A). Compared with the control, the three treatments remarkably reduced the number of fungal OTUs in the soil (*p* < 0.05) (Figure 4B).

For the bacterial communities, the PCoA revealed that the samples from the same treatment were grouped together, and the samples from the four treatments could be clearly distinguished (Figure 4C); however, for the fungal communities, there was little difference between the four treatments (Figure 4D).

### 3.6. Compositions of Bacterial and Fungal Communities in Soil

The Miseq sequencing data that were classified at the 97% similarity level included 40 bacterial phyla and 7 fungal phyla. For the bacterial phylum community, there were 10 phyla with a relative abundance > 1%. Compared with the CK, the potato onion, soybean, and wheat rotation treatments showed a lower relative abundance of Proteobacteria and Firmicutes (*p* < 0.05), and a higher relative abundance of Chloroflexi (*p* < 0.05); the relative abundance of Acidobacteria in the potato onion and wheat treatments was significantly higher than the control (*p* < 0.05) (Figure 5A and Appendix A). For the fungal phylum community, there were three phyla with a relative abundance > 0.1%. The dominant phyla were Ascomycota and Basidiomycota, with a total abundance > 85%. The relative abundance of Basidiomycota in the three rotation treatments was lower than in the control, albeit not significantly (*p* > 0.05) (Figure 5B).

Among all the samples, the potato onion rotation had the highest number of unique OTUs (1572), and the control had the lowest number of unique OTUs (350) in bacteria (Figure 5C). The control had the highest number of unique OTUs (216) and bean rotation had the lowest number of unique OTUs (78) in fungal (Figure 5D).

As for classes level, 106 bacterial and 21 fungal taxa were detected. For the bacterial class communities, there were 16 classes with a relative abundance > 1%. Alphaproteobacteria and Saccharimonadia were the highest in the control, while Deltaproteobacteria and Acidimicrobiia were lower in the control than in the other treatments. The relative abundance of Actinobacteria was lower in the bean and wheat treatments than in the control. Compared with the control, bean rotation had a higher relative abundance of Thermoleophilia, while that of potato onion rotation was lower (*p* < 0.05) (Figure 6A and Appendix A). For the fungal class community, there were five classes with a relative abundance > 1%. The abundance of Sordariomycetes was higher in the three treatments than in the control, while those of Agaricomycetes, Eurotiomycetes, and Leotiomycetes were lower than in the control (*p* > 0.05) (Figure 6B and Appendix A).

In total, 783 bacterial genera and 186 fungal genera were classified. For the bacterial genus community, there were 19 genera with a relative abundance > 0.5%. The dominant genera were *Sphingomonas*, *Gemmatimonas*, *RB41*, *Pirellula*, *Lysobacter*, and *Gemmata* spp.; *Sphingomonas*, *Gemmatimonas*, and *Rhodanobacter* spp. were higher in the control than in the other treatments, while *Subgroup_10* and *Arenimonas* spp. were lowest in the control. The potato onion and wheat rotation had a higher *RB41* relative abundance than the control; the bean and potato rotation had a higher *Pirellula* spp. relative abundance than the control. Compared with the control, the bean rotation had a higher *Lysobacter* spp. and lower *Bryobacter* and *Luteimonas* spp. relative abundance. The wheat rotation increase the relative abundance of *Chthoniobacter* spp. and reduced *Nocardioides* spp. and *Luteimonas* spp. compared with the control (*p* < 0.05) (Table 2). For the fungal genus community, there were 10 genera with a relative abundance > 0.5%. The dominant genera were *Fusarium* and *Cladorrhinum* spp. Compared with the control, the bean rotation increased the relative abundance of *Cladorrhinum* significantly; the potato onion rotation and wheat rotation reduced the relative abundance of *Conocybe* remarkably (*p* < 0.05) (Table 3).

### 3.7. Relationships between Soil Microbial Communities and Soil Chemical Properties

A redundancy analysis (RDA) was used to analyze the relationship between the changes in the bacterial and fungal community structures and the environmental factors among the different treatments. The Mantel test results showed that changes in bacterial community structure were associated with the soil pH (r = 0.4342, *p* = 0.012), EC (r = 0.6642, *p* = 0.001), available K (r = 0.5469, *p* = 0.001), and available N (r = 0.4320, *p* = 0.011) (Figure 7A), while the soil pH (r = 0.2979, *p* = 0.035) and available K (r = 0.3089, *p* = 0.013) were the dominant factors driving the changes in soil fungal community structure (Figure 7B) (Appendix A).

### 3.8. Microbial Ecological Guilds in Soil

There are six types of primary bacterial functional layers, including metabolism, environmental information processing, genetic information processing, cellular processes, human diseases and organismal systems.

In the primary bacterial metabolic pathway, the control had the lowest relative abundance of human diseases. The relative abundance of metabolism was lowest in the bean rotation, while organismal systems and genetic information processing were higher than in the control. The bean and wheat rotations increased the relative abundance of cellular processes remarkably compared with the control (*p* < 0.05) (Figure 8).

In the fungi, seven primary functional layers were analyzed, including pathotroph-symbiotrophic fungi, pathotroph-saprotrophic fungi, saprotrophic-symbiotrophic fungi, symbiotrophic fungi, pathotroph, saprotrophic fungi and pathtroph-saprotrophic-symbiotrophic fungi.

In the primary fungal ecological function, the relative abundances of pathotroph and pathoph-saprotroph-symbiotroph were lower in the potato onion and wheat rotations than in the control; compared with the control, the bean rotation had a higher relative abundance of pathotroph-saprotroph, the potato onion rotation had a higher relative abundance of saprotroph, and wheat rotation had more symbiotroph (*p* < 0.05) (Figure 9).

### 3.9. Correlation Analysis between Resting Spores and Soil Microorganisms

A Pearson correlation analysis was conducted between the OTUs with a relative abundance > 0.5% and the resting spores of *P. brassicae* (*p* < 0.05).

There were 11 bacterial OTUs related to the resting spores. OTU 1 (*Sphingomonas*), OTU 3 (*Sphingomonas*), OTU 17 (Sphingomonadaceae), OTU 25 (Subgroup_6), OTU 63 (*Gemmatimonas*), and OTU 92 (Gemmatimonadaceae) had an extremely significantly positive correlation with the resting spores (*p* < 0.01), while OTU 8 (Sphingomonas) and OTU 11 (Gemmatimonadaceae) were positively correlated with the resting spores. OTU 39(Gemmatimonadaceae) was extremely negatively correlated with the resting spores (*p* < 0.05). OTU 9(*Lysobacter*) had significantly negative correlation with resting spores (*p* < 0.01) (Table 4).

For the fungi, four OTUs had a correlation with the resting spores. OTU 9(*Acephala*), OTU 15 (*Penicillium*), and OTU 27 (GS29) were extremely significantly positively correlated with the resting spores (*p* < 0.01). OTU 31 (Ascomycota) had an obviously positive correlation with the resting spores (*p* < 0.05) (Table 5).

## 4. Discussion

### 4.1. Effects of Rotation on Growth of Chinese Cabbage

Previous studies have shown that rotation can promote plant growth and increase yield [21]. Our study found that, compared with the garlic rotation, the onion and wheat rotations increased the yield of the Chinese cabbage under the experimental conditions for two consecutive years. On the one hand, this was possibly related to the allelopathy of the preceding crops; for example, the allelochemicals secreted by wild mustard can promote the growth and increase the yield of cauliflower [22]. Root exudates also play a certain role in the growth of plants. The root exudates of potato onions promote the growth and improve the photosynthesis of tomatoes [23]. However, during the cultivation of cucumber/tomato, buckwheat/corn and sorghum/sesame in the same environment, the respective plants inhibit each other. On the other hand, environmental conditions more conductive to the growth of Chinese cabbage are created after preceding crops are planted and harvested. The growth of eggplant was significantly better when it was intercropped with corn than when it was alone [24]. Potato onion and wheat rotations can reduce the stress of *P. brassicae* on Chinese cabbage and alleviate the degree of hindrance of plant water and nutrient absorption, thereby promoting the growth of Chinese cabbage and increasing its yield.

### 4.2. Effects of Rotation on the Soil’s Chemical Properties

Rotation can improve soil nutrients and increase the effectiveness of nutrient utilization, thereby promoting the growth of crops [25]. Different crops utilize different soil nutrients. Our study found that the soybean, potato onion, and wheat rotation treatments significantly reduced the EC value and potassium content, and increased the pH value of the soil. The content of potassium increased with the aggravation of clubroot, probably because the *P. brassicae* destroyed the cortexes and columns of the Chinese cabbage, hindering the absorption and utilization of nutrients in the soil by the roots. The suitable pH value for the Chinese cabbage was 6.5–7.0; rotation improved the soil pH, promoting the growth of the Chinese cabbage.

### 4.3. Effects of Rotation on Soil Microbial Communities

Soil microorganisms are important parts of soil fertility, participating in organic carbon decomposition, humus formation, and soil nutrient transformation and cycling. Higher plants can improve soil diversity by increasing soil carbon source diversity [26]. Generally, soil fertility is positively correlated with less fungal and more bacterial content in soil microbial community structures [27]. Soil community diversity comprehensively reflects the richness and evenness of communities; community richness is a measure of alpha diversity, representing the number of species in an ecosystem. Our study found that preceding soybean, potato onion, and wheat could increase the richness and diversity of soil bacteria and decrease the richness and diversity of fungi. Preceding crops provide nutrients to soil microorganisms through root exudates and plant residues [28]. Crop rotation can increase crop diversity and organic carbon input to a certain extent. Diversified carbon sources are able to support the growth and reproduction of diverse microbial groups, increasing soil bacterial diversity [29]. The abundances of potential pathogens and antagonistic microorganisms are lower in rotation soils, while those of potentially plant-promoting microorganisms are higher [30]. Xiong et al., sequenced the soil of long-term monoculture of vanilla, and found that the bacteria Actinomycota and Firmicutes were more abundant in disease-suppressive soil [31]. Through the comparison of species differences, our study found that the rotations of potato onion, soybean and wheat significantly reduced the relative abundance of Proteobacteria and Firmicutes. The comparison of the differences in the soil bacterial communities found that, compared with the control, the *Gemmatimonas* spp. remarkably decreased in the three treatments during the harvesting of the preceding crops. *Gemmatimonas* spp. is an oligotrophic flora that can promote the growth of and reduce disease in plants. Its lower abundance in the treatments was possibly due to the presence of richer soil nutrients during the cultivation of Chinese cabbage [32].

### 4.4. Effects of Rotation on the Clubroot of Chinese Cabbage

The occurrence of soil-borne diseases is a comprehensive manifestation of the interaction between crops and soil systems, involving changes in soil microbial community diversity, the accumulation of allelochemicals, imbalances in soil nutrients, and the degradation of soil physicochemical properties. These changes influence each other and work together, eventually leading to the occurrence of soil-borne diseases [33,34]. This experiment found that compared with the control, the three rotations all significantly reduced the content of the *P. brassicae* resting spores in the soil, as well as the incidence and disease index of the Chinese cabbage, during the two years. This was, probably because the physicochemical properties of the soil were altered by the planting of preceding crops, thereby reducing the incidence of clubroot. Soil moisture has a significant and persistent effect on clubroot pathogenesis. Under favorable temperature conditions, a soil moisture level of 60–70% is the most favorable for clubroot, and even with lower soil moisture, low levels of infection can occur. The incidence of clubroot in the field test was more significant because of the large amount of precipitation and high soil moisture. Studies have shown that rainfall is positively correlated with the severity of clubroot in Brassica vegetables, such as radish [35]. The suitable pH value for the survival of *P. brassicae* is 5.4–6.7 [36]. In neutral soil pH, the inhibition of spore germination is the main reason for the inhibition of clubroot [37]. Therefore, rotation increases the suitable pH value of *P. brassicae*, promoting its germination in soil, which leads to the death of zoospores without suitable live hosts, thereby reducing the content of soil *P. brassicae* and the occurrence of diseases.

The root exudates of non-host crops induce the germination of resting spores, resulting in the death of zoospores because of the absence of host plants. Many compounds in root exudates, such as carbohydrates, nucleosides, auxins, enzymes, and phenols, are capable of providing effective carbon and nitrogen sources for pathogens. Nitrogen, sodium and manganese can inhibit, while boron can stimulate the germination of resting spores. Sterols, especially sterols secreted by roots, can play a certain role in the germination of *P. brassicae* resting spores. Some hydrolyzates, such as trans-2-hydroxy-3-butene glucosinolate and 4-hydroxy-3-indolemethyl glucosinolate, can also stimulate the germination of resting spores [38]. The quantity and quality of root exudates depend on the plant species, the age of individual plants, and external factors, such as biotic or abiotic stress sources. Environmental factors, such as temperature, light, and soil moisture, also regulate the processes of root exudation [39]; for example, the root exudates of radish stimulate the germination of resting spores [40], while the root exudates of potato onions have an inhibitory effect on the germination of resting spores [41]. The next step should be to analyze the composition of the root exudates of previous crops and study their effect on the germination of dormant spores. A component analysis of the root exudates of preceding crops and their effects on the germination of resting spores should be carried out in further experiments.

The preceding crops caused changes in the soil microorganisms, which altered the ecological environment of the soil, enhancing the resistance of the plants to pathogenic bacteria [42,43]. Through high-throughput sequencing, it was found that the three preceding crops increased the richness and diversity of the soil bacteria and decreased the richness and diversity of the fungi after harvesting. The proteins related to the plant disease process are differentially expressed when plants are infected by pathogenic bacteria, which improves the resistance of crops to pathogenic bacteria [44]. Through the correlation analysis between the content of the *P. brassicae* resting spores and the soil microorganisms, it was found that the bacteria OTU 9 and OTU 39 were significantly negatively correlated with the content of the resting spores. OTU 9 belongs to *Lysobacter* whose relative abundance in the control was significantly lower than in the other treatments. *Lysobacter* can produce a large number of active secondary metabolites and extracellular enzymes, which can act on the cell walls of pathogenic bacteria to dissolve or inhibit them [45]. *Lysobacter capsici* AZ78 has toxic activity against the sporangia of *Phytophthora infestans* and *Plasmopara viticola* [46]; the metabolites of *Lysobacter capsici* YS1215 have root-knot nematode active substances [45]. Recent studies have demonstrated that *Lysobacter* can produce antifungal compounds that inhibit the reproduction of *P. brassicae* resting spores and alleviate clubroot disease [47]. OTU 1, OTU 3, OTU 8, OTU 17, and OTU 22 which all belong to *Sphingomonas* spp., were significantly positively correlated with the content of the resting spores, and their numbers were significantly decreased after the planting of the preceding crops. *Sphingomonas* is often used to protect plants against microbial disease as a biological control agent through substrate competition. This indicates that some species of *Sphingomonas* spp. might have a synergistic effect with *P. brassicae*; this needs to be verified in later studies. *Sphingomonas* strains protected Arabidopsis thaliana against leaf pathogenic *Pseudomonas syringae* in a controlled model system [48]. OTU 39 and OTU 92 belong to the Gemmatimonadaceae family, the genus was unclassified.

Among the fungi, OTU9, OTU15 and OTU 31 belonged to Ascomycota. Ascomycetes are mainly saprophytes, which can decompose organic matter into nutrients that can be absorbed by crops and decompose cellulose and lignin in soil [49]. OTU 27 belongs to Basidiomycota, which easily decomposes lignin with a high carbon-to-nitrogen ratio; increases in lignin content increase its relative abundance [50]. Based on this, we speculated that the Ascomycota and Basidiomycota were probably involved in promoting the germination of the resting spores of *P. brassicae*.

## 5. Conclusions

Overall, different rotations have different effects on plants and soil. Compared with the control, the rotation of the potato onion, bean and wheat treatments significantly reduced the clubroot disease index, promoted the growth of the Chinese cabbage, increased the yield of the Chinese cabbage, and improved soil chemistry. In addition, rotation decreased the content of the pathogenic bacteria in the soil, increased the richness and diversity of bacteria, and decreased the richness and diversity of fungi. This experiment shows that the rotation of non-cruciferous previous crops and cruciferous crops is an important strategy to improve the sustainable development of agriculture. Additional work is needed to explore the specific disease-inhibiting mechanism behind and the long-term effects of rotation on soil pathogens and other microorganisms.

## Figures and Tables

**Figure 1 microorganisms-10-00799-f001:**
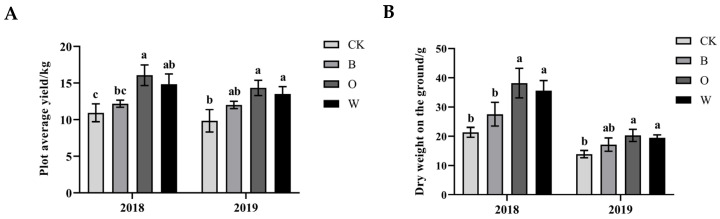
Effects of different preceding crops on yield (**A**) and dry weight on the ground (**B**) of Chinese cabbage. CK represents Chinese cabbage continuous cropping; B, O, and W represent bean, potato onion, and wheat rotation respectively. Different letters are significantly different; the results are for the statistical analysis within each year (*p* < 0.05, Tukey’s HSD test).

**Figure 2 microorganisms-10-00799-f002:**
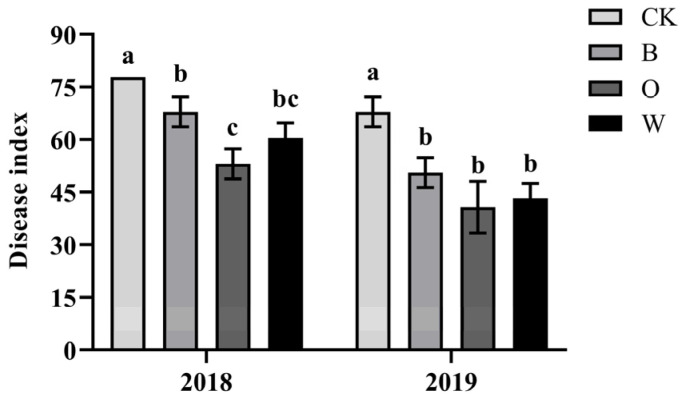
Disease index of Chinese cabbage planting for 40 days. CK represents Chinese cabbage continuous cropping; B, O, and W represent bean, potato onion and wheat rotation respectively. Different letters are significantly different; the results are for the statistical analysis within each year (*p* < 0.05, Tukey’s HSD test).

**Figure 3 microorganisms-10-00799-f003:**
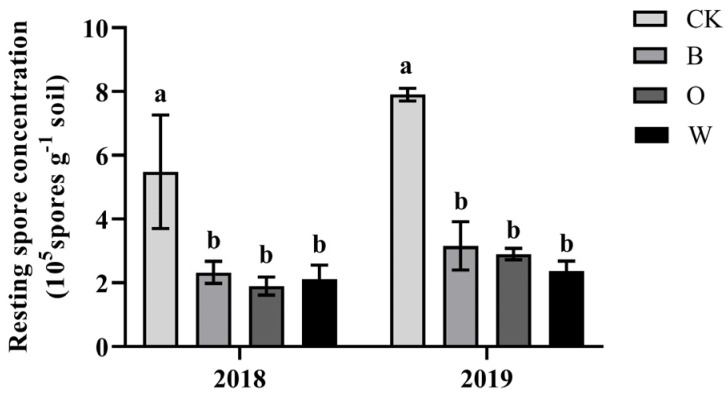
Concentrations of resting spores of *plasmodiophora brassicae* after preceding crops were harvested. CK represents Chinese cabbage continuous cropping, B, O, and W represent bean, potato onion, and wheat rotation respectively. Different letters are significantly different, the results are for the statistics analysis within each year (*p* < 0.05, Tukey’s HSD test).

**Figure 4 microorganisms-10-00799-f004:**
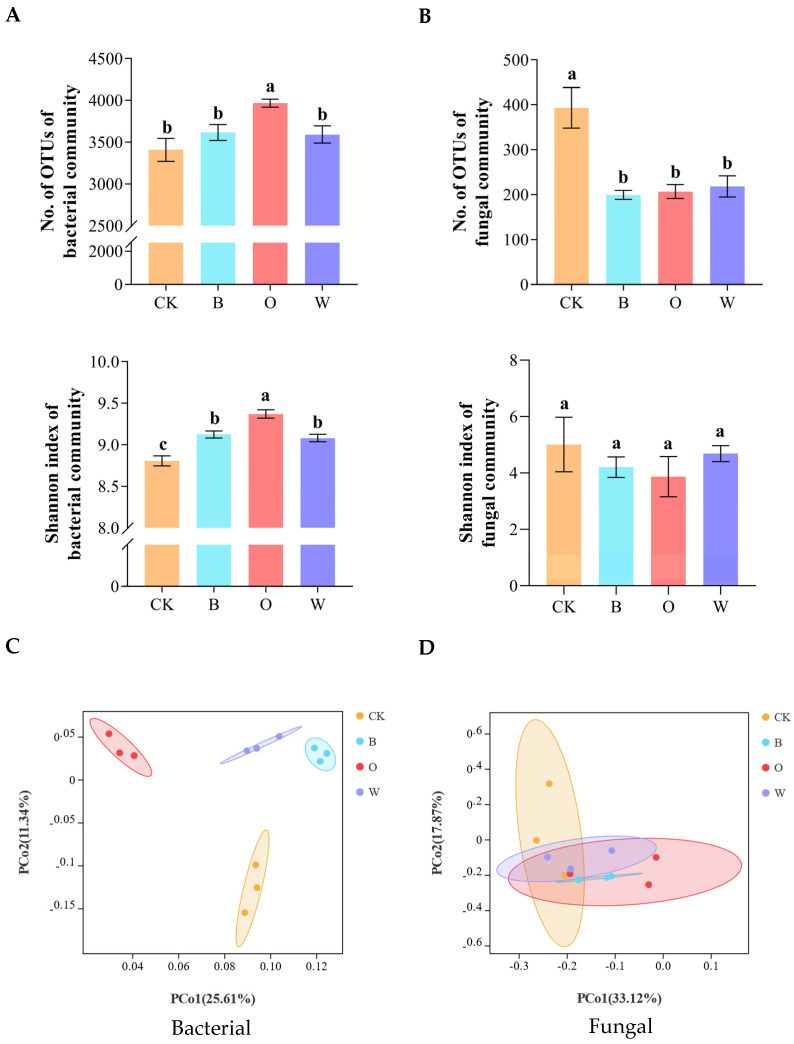
Alpha (**A**) and beta (**C**) diversities of soil bacterial and alpha (**B**) and beta (**D**) diversities of soil fungal communities. Beta diversities based on Bray Curtis distance dissimilarity were visualized by principal component analyses. OTUs were delineated at 97% sequence similarity. CK represents Chinese cabbage continuous cropping; B, O, and W represent bean, potato onion and wheat rotation respectively. Different letters are significantly different (*p* < 0.05, Tukey’s HSD test).

**Figure 5 microorganisms-10-00799-f005:**
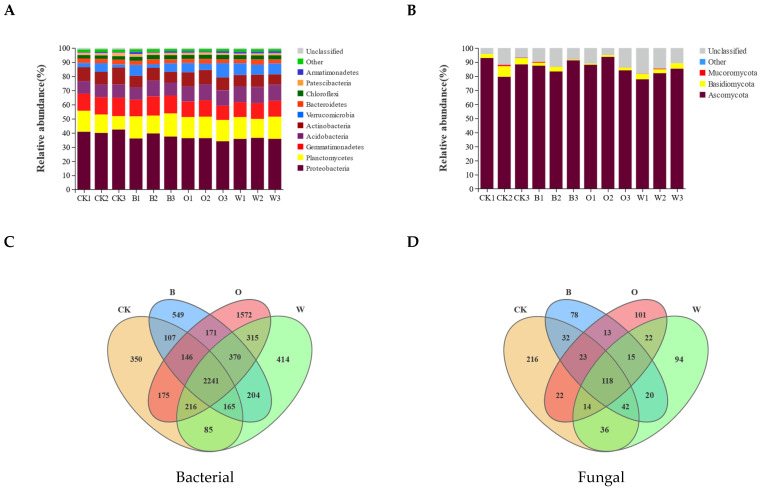
Relative abundances of bacterial (**A**) and fungal (**B**) phyla and Venn diagram analyses of bacterial (**C**) and fungi (**D**) in soil of different treatments. Bacterial and fungal phyla with average relative abundance > 1% are shown and do not contain unclassified taxa. Data are represented as the means of three independent replicates. Venn diagram (**C**,**D**) demonstrates the numbers of shared and unique observed OTUs at 97% similarity among treatments.

**Figure 6 microorganisms-10-00799-f006:**
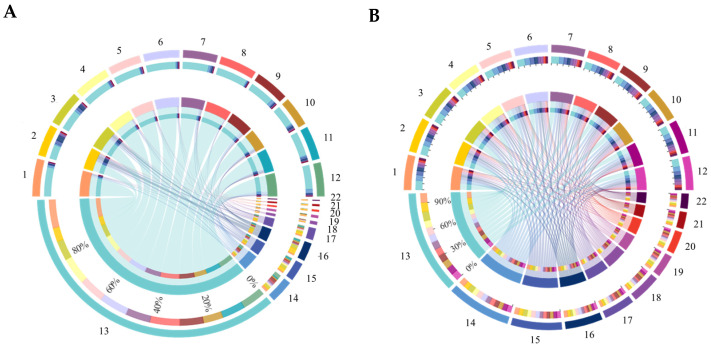
Community compositions of bacterial (**A**) and fungal (**B**) classes in soil of different treatments. For figure (**A**), numbers represent: 1. CK1, 2. CK2, 3. CK3, 4. B1, 5. B2 6. B3, 7. O1, 8. O2, 9. O3, 10. W1, 11. W2, 12. W 3, 13. Alphaproteobacteria, 14. Gammaproteobacteria, 15. Gemmatimonadetes, 16. Phycisphaerae, 17. Planctomycetacia, 18. Verrucomicrobiae, 19. Actinobacteria, 20. Subgroup_6, 21. Blastocatellia_Subgroup_4, and 22. Bacteroidia. For figure (**B**), numbers represent: 1. CK1, 2. CK2, 3. CK3, 4. B1, 5. B2 6. B3, 7. O1, 8. O2, 9. O3, 10. W1, 11. W2, 12. W3, 13. Sordariomycetes, 14. Dothideomycetes, 15. Eurotiomycetes, 16. Leotiomycetes, 17. Agaricomycetes, 18. Pezizomycetes, 19. Tremellomycetes, 20. Saccharomycetes, 21. Mucoromycetes, and 22. Spizcllomycetes.

**Figure 7 microorganisms-10-00799-f007:**
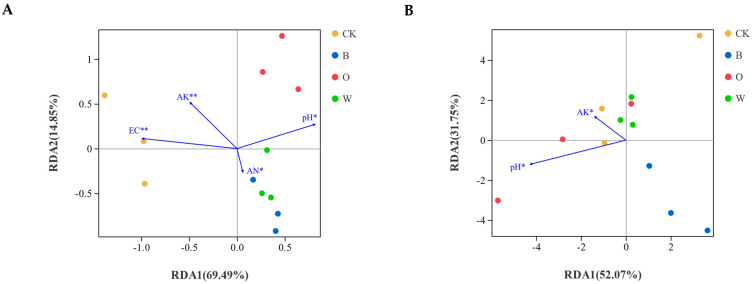
Redundancy analysis (RDA) of soil total bacterial (**A**) and fungal (**B**) community structures during the harvesting of the preceding crops. The environmental variables with statistical significance are presented by arrows. AK and AN indicate available K and available N, respectively CK represents Chinese cabbage continuous cropping, B, O, and W represent bean, potato onion, and wheat rotation, respectively. * *p* < 0.05; ** *p* < 0.01.

**Figure 8 microorganisms-10-00799-f008:**
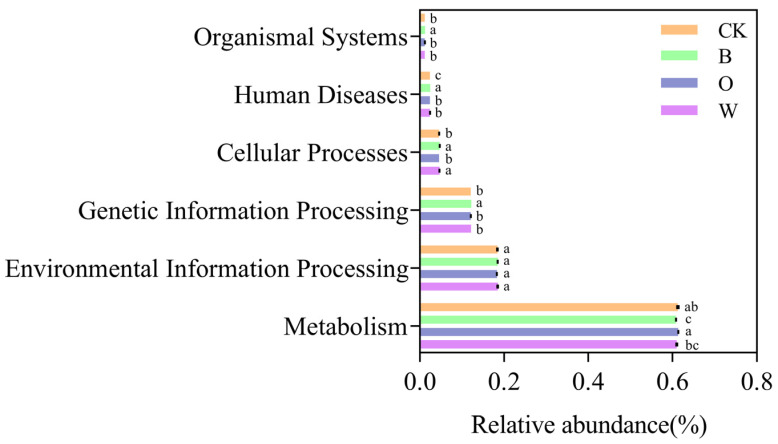
Soil bacterial function prediction of soil in different treatments (Hierarchy Level 1). CK represents Chinese cabbage continuous cropping; B, O, and W represents bean, potato onion and wheat rotation, respectively. Different letters are significantly different (*p* < 0.05, Tukey’s HSD test).

**Figure 9 microorganisms-10-00799-f009:**
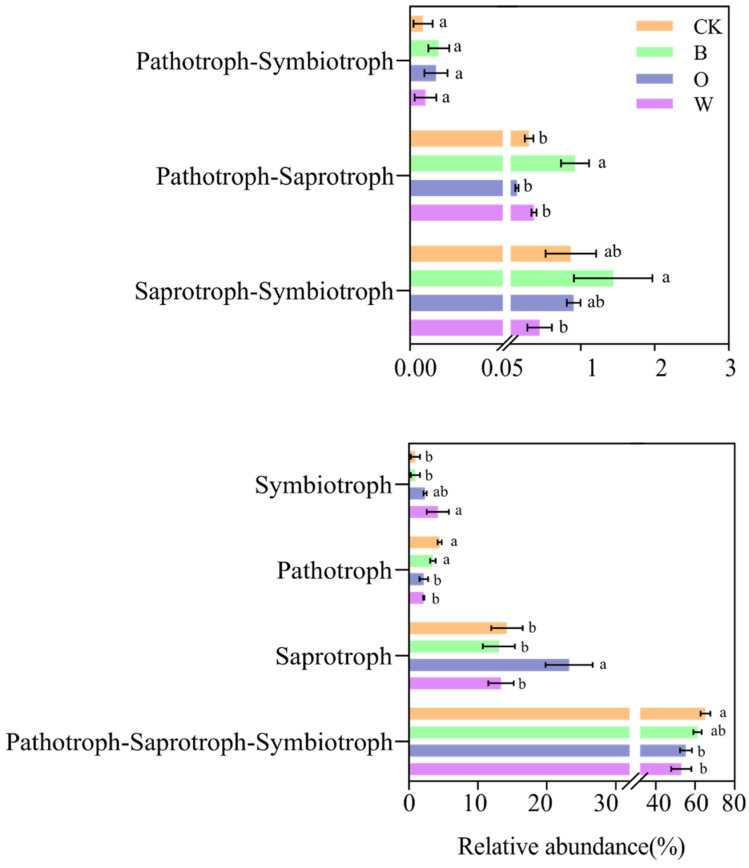
Soil fungal function prediction of soil in different treatments (Hierarchy Level 1). CK represents Chinese cabbage continuous cropping, B, O, and W represent bean, potato onion, and wheat rotation respectively. Different letters are significantly different (*p* < 0.05, Tukey’s HSD test).

**Table 1 microorganisms-10-00799-t001:** Effects of different preceding crops on soil chemical properties.

		EC (mS/cm)	pH	SOM (%)	AN (mg/kg)	AP (mg/kg)	AK (mg/kg)	TN (g/kg)
2018	CK	0.31 ± 0.02a	6.32 ± 0.04c	3.43 ± 0.23a	116.74 ± 3.71a	107.67 ± 18.06a	652.50 ± 32.50a	2.63 ± 0.16b
	B	0.10 ± 0.01c	6.48 ± 0.06b	3.42 ± 0.21a	117.39 ± 6.13a	115.41 ± 1.25a	611.67 ± 30.55a	2.84 ± 0.20b
	O	0.12 ± 0.01bc	6.73 ± 0.06a	3.10 ± 0.47a	110.93 ± 2.84a	107.86 ± 6.88a	634.17 ± 23.23a	2.09 ± 0.14c
	W	0.13 ± 0.01b	6.76 ± 0.03a	3.36 ± 0.39a	132.00 ± 17.52a	117.51 ± 8.74a	535.83 ± 18.09b	4.24 ± 0.15a
2019	CK	0.22 ± 0.02a	6.56 ± 0.06c	3.93 ± 0.34a	148.59 ± 19.12a	90.45 ± 12.75a	569.33 ± 12.86a	1.13 ± 0.12a
	B	0.13 ± 0.01b	6.55 ± 0.06c	4.12 ± 0.50a	157.82 ± 3.33a	92.48 ± 5.57a	401.33 ± 4.62c	1.08 ± 0.04a
	O	0.14 ± 0.01b	7.08 ± 0.02a	3.96 ± 0.43a	160.97 ± 5.37a	91.08 ± 3.77a	528.00 ± 6.93b	1.09 ± 0.02a
	W	0.12 ± 0.00b	6.67 ± 0.01b	4.25 ± 0.18a	154.84 ± 2.57a	96.26 ± 7.48a	396.00 ± 17.44c	1.12 ± 0.04a

CK represents Chinese cabbage continuous cropping; B, O, and W represent bean, potato onion and wheat rotation, respectively. Different letters are significantly different (*p* < 0.05, Tukey’s HSD test).

**Table 2 microorganisms-10-00799-t002:** Relative abundances of dominant bacterial genera in soil.

	CK	B	O	W
*Sphingomonas*	15.99 ± 0.44a	12.60 ± 0.37b	12.53 ± 0.41b	13.02 ± 0.08b
*Gemmatimonas*	4.42 ± 0.30a	3.48 ± 0.18b	3.26 ± 0.210b	3.19 ± 0.07b
*Candidatus_Udaeobacter*	2.52 ± 1.50a	3.27 ± 1.50a	3.94 ± 1.91a	4.76 ± 0.40a
*RB41*	1.41 ± 0.12c	0.45 ± 0.25bc	1.91 ± 0.19ab	2.17 ± 0.18a
*Pirellula*	0.86 ± 0.31b	1.84 ± 0.53a	2.02 ± 0.40a	1.69 ± 0.07ab
*Lysobacter*	1.68 ± 0.05bc	2.23 ± 0.19a	1.52 ± 0.13c	1.84 ± 0.07b
*Gemmata*	0.78 ± 0.23a	0.94 ± 0.31a	1.29 ± 0.35a	1.09 ± 0.11a
*Streptomyces*	0.42 ± 0.09a	0.20 ± 0.06a	0.21 ± 0.13a	0.19 ± 0.04a
*Marmoricola*	1.08 ± 0.34a	0.63 ± 0.09a	0.83 ± 0.09a	0.65 ± 0.21a
*Nocardioides*	0.92 ± 0.30a	0.50 ± 0.01ab	0.72 ± 0.06ab	0.61 ± 0.07b
*Chthoniobacter*	0.31 ± 0.16b	0.61 ± 0.29ab	0.74 ± 0.25ab	0.90 ± 0.06a
*Flavisolibacter*	0.55 ± 0.10a	0.59 ± 0.05a	0.60 ± 0.09a	0.61 ± 0.10a
*Haliangium*	0.51 ± 0.02a	0.69 ± 0.05a	0.63 ± 0.09a	0.50 ± 0.12a
*Rhodanobacter*	0.93 ± 0.17a	0.44 ± 0.03b	0.38 ± 0.02b	0.55 ± 0.09b
*Bryobacter*	0.65 ± 0.10a	0.39 ± 0.01b	0.65 ± 0.05a	0.59 ± 0.10ab
*Ellin6067*	0.51 ± 0.09a	0.52 ± 0.12a	0.57 ± 0.06a	0.50 ± 0.07a
*Subgroup_10*	0.28 ± 0.00b	0.48 ± 0.10a	0.53 ± 0.05a	0.58 ± 0.08a
*Luteimonas*	0.63 ± 0.10a	0.36 ± 0.03b	0.52 ± 0.03a	0.31 ± 0.03b
*Arenimonas*	0.24 ± 0.03c	0.60 ± 0.01a	0.42 ± 0.05b	0.50 ± 0.07ab

Values (mean ± standard error) in the same row followed by different letters are significantly different at the 0.05 probability level, according to Tukey’s HSD test. Bacterial genera with average relative abundance > 0.5% are shown and do not contain unclassified taxa. Different letters are significantly different (*p* < 0.05, Tukey’s HSD test).

**Table 3 microorganisms-10-00799-t003:** Relative abundances of dominant fungal genus in soil.

	CK	B	O	W
*Fusarium*	30.33 ± 15.22a	30.06 ± 1.92a	38.96 ± 16.44a	31.36 ± 4.80a
*Cladorrhinum*	3.59 ± 3.54b	18.61 ± 9.49a	1.84 ± 0.44b	3.68 ± 2.43b
*Acephala*	6.09 ± 4.47a	0.30 ± 0.50a	1.62 ± 1.08a	2.44 ± 1.14a
*Penicillium*	2.01 ± 1.09a	0.31 ± 0.230a	1.78 ± 0.94a	0.48 ± 0.69a
*Gibberella*	2.01 ± 1.28a	1.81 ± 1.53a	2.07 ± 0.22a	0.86 ± 0.13a
*Talaromyces*	1.47 ± 1.20a	0.27 ± 0.33a	0.29 ± 0.28a	1.71 ± 1.52a
*Conocybe*	0.15 ± 0.04a	0.27 ± 0.13ab	0.14 ± 0.08b	1.06 ± 0.62b
*Gliocladium*	0.18 ± 0.19a	0.90 ± 0.68a	0.62 ± 0.52a	0.90 ± 0.40a
*Alternaria*	2.23 ± 1.47a	0.59 ± 0.43a	3.27 ± 3.42a	0.90 ± 0.89a
*Aspergillus*	1.57 ± 0.96a	0.89 ± 0.75a	0.13 ± 0.10a	0.63 ± 0.49a

Different letters are significantly different (*p* < 0.05, Tukey’s HSD test).

**Table 4 microorganisms-10-00799-t004:** Correlation analysis between resting spores and soil bacterial OTUs.

Bacterial OTUs (Relative Abundance in Soil %)	Correlation Coefficient with Resting Spores	Phylum	Family	Genus
OTU 1 (8.591)	0.765 **	Proteobacteria	Sphingomonadaceae	*Sphingomonas*
OTU 3 (2.225)	0.755 **	Proteobacteria	Sphingomonadaceae	*Sphingomonas*
OTU 8 (1.148)	0.687 *	Proteobacteria	Sphingomonadaceae	*Sphingomonas*
OTU 9 (0.972)	−0.684 *	Proteobacteria	Xanthomonadaceae	*Lysobacter*
OTU 11 (0.795)	0.697 *	Gemmatimonadetes	Gemmatimonadaceae	Unclassified
OTU 17 (0.608)	0.826 **	Proteobacteria	Sphingomonadaceae	*Sphingomonas*
OTU 22 (0.433)	0.626 *	Proteobacteria	Sphingomonadaceae	*Sphingomonas*
OTU 25 (0.428)	0.754 **	Acidobacteria	Unclassified	Unclassified
OTU 39 (0.590)	−0.717 **	Gemmatimonadetes	Gemmatimonadaceae	Unclassified
OTU 63 (0.320)	0.766 **	Gemmatimonadetes	Gemmatimonadaceae	*Gemmatimonas*
OTU 92 (0.191)	0.832 **	Gemmatimonadetes	Gemmatimonadaceae	Unclassified

“*” and “**” represent significant differences at the 0.05 and 0.01 probability level respectively according to Pearson correlation analysis. “−” represents negative correlation.

**Table 5 microorganisms-10-00799-t005:** Correlation analysis between resting spores and soil fungal OTUs.

Fungal OTUs (Relative Abundance in Soil%)	Correlation Coefficient with Resting Spores	Phylum	Family	Genus
OTU 9 (2.609)	0.720 **	Ascomycota	Vibrisseaceae	*Acephala*
OTU 15 (0.571)	0.907 **	Ascomycota	Trichocomaceae	*Penicillium*
OTU 27 (0.919)	0.822 **	Basidiomycota	Agaricomycetes GS29	Unclassified
OTU 31 (0.742)	0.616 *	Ascomycota	Unclassified	Unclassified

“*” and “**” represent significant differences at the 0.05 and 0.01 probability levels, respectively according to the Pearson correlation analysis.

## Data Availability

Data are available by contacting the authors.

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
