# Peer review of "Three Preceding Crops Increased the Yield of and Inhibited Clubroot Disease in Continuously Monocropped Chinese Cabbage by Regulating the Soil Properties and Rhizosphere Microbial Community"

_microorganisms, 2022, doi:10.3390/microorganisms10040799_

Round 1

Reviewer 1 Report

The title of the work does not describe the scientific content of the study well, it should be improved.

There are spelling errors. For example, in Figure 2, in the graphical name of the YY axis, and in section 3.8 the name of soil fungal “…Pathotroph-Saprotrophic-Symbiotrophic”.

In the material and methods, there are sections with the same title (points 2.1 and 2.2).

In point 2.5, the authors do not refer all parameters that were evaluated and are presented in Table 1 (i.e. EC methodology).

The authors must insert the meaning of acronyms, the first time they appear in the text.

In Figures 1 and 2, it is confusing to use the same significance letters for the two years, since the results are for the statistic analysis within each year. The authors should clarify this in the Figures.

The interpretation of the results is not consistent with what is presented in Figures.

The results interpretation should be revised. For instance in section 3.1, the authors wrongly refered that “… in 2019 in higher than 2018, …” in the section 3.6, line 223 “… in bacterial.”, and in section 3.8 the discussion of ANOVA data from Figure 9 for the PSS assay are not correct in the text “…. Pathotroph-Saprotroph-Symbiotroph were higher in potato onion and wheat rotations than control, …”.

The signicant figures should be checked, because standard deviations are presented in too much detail.

In the analysis of statistical data, for example, highlighting correlations lower than 0.9 does not bring relevance to the work (Table 4).

The graphics quality needs to be improved.

Author Response

Response to Reviewer 1 Comments

Dear Reviewer,

Here is our revised manuscript (Manuscript ID: microorganisms-1652351) for “Microorganisms”. Response to the comments:

Reviewer reports:

Point 1: The title of the work does not describe the scientific content of the study well, it should be improved.

Response 1: Thank you very much for your professional advice. We changed the title to “Three Preceding Crops Increased the Yield, Inhibited Clubroot Disease of Continuously Monocropped Chinese Cabbage by Regulating the Soil Properties and Rhizosphere Microbial Community” based on your suggestion.

Point 2: There are spelling errors. For example, in Figure 2, in the graphical name of the YY axis, and in section 3.8 the name of soil fungal “…Pathotroph-Saprotrophic-Symbiotrophic”.

Response 2: Thank you very much for your kind advice. We also modified the name of Y axis in Figure 2 based on your suggestions, we changed “Disease incidence” to “Disease index”. In 3.8, we modified the description in the results.

Point 3: In the material and methods, there are sections with the same title (points 2.1 and 2.2).

Response 3: We are deeply sorry for our mistake. The title of point 2.2 was changed to “Experimental Design” according to your kind suggestion.

Point 4: In point 2.5, the authors do not refer all parameters that were evaluated and are presented in Table 1 (i.e. EC methodology).

Response 4: We have added EC methodology in line 115.

Point 5: The authors must insert the meaning of acronyms, the first time they appear in the text.

Response 5: Thank you very much for your responsible comments. We explained what EC, SOM, AN, AP, AK and TN mean when they first appear in the text in lines 80-83.

Point 6: In Figures 1 and 2, it is confusing to use the same significance letters for the two years, since the results are for the statistic analysis within each year. The authors should clarify this in the Figures.

Response 6: According to your suggestion, we have added the explanation for significance letters. Different letters are significantly different, the results are for the statistic analysis within each year.

Point 7: The interpretation of the results is not consistent with what is presented in Figures.

The results interpretation should be revised. For instance in section 3.1, the authors wrongly refered that “… in 2019 in higher than 2018, …” in the section 3.6, line 223 “… in bacterial.”, and in section 3.8 the discussion of ANOVA data from Figure 9 for the PSS assay are not correct in the text “…. Pathotroph-Saprotroph-Symbiotroph were higher in potato onion and wheat rotations than control, …”.

Response 7: Thank you very much for your kind advice. We have revised the results analysis that do not present in the figures including these two points.

Point 8: The signicant figures should be checked, because standard deviations are presented in too much detail.

Response 8: We modified the data (including standard deviations) in tables 2-3 to two decimal places reserved.

Point 9: In the analysis of statistical data, for example, highlighting correlations lower than 0.9 does not bring relevance to the work (Table 4).

Response 9: Thank you very much for your responsible comments. The statistical data of table 4 was analyzed through The Pearson correlation coefficient represents the correlation between the module and the sample, P value represents the significant correlation coefficient.

Point 10: The graphics quality needs to be improved.

Response 10: We have increased the resolution of figures 1-8, and changed color in figure 4.

In addition, regarding your kind suggestion that English language and style required extensive editing, our reply is that the manuscript has edited by a professional scientific editing service. Thank you again for your kind suggestion. If there is any shortage, please inform us!

Finally, the manuscript was revised based on the comments of reviewers and editors. All changes in the manuscript use tracking marks. If there is any shortage, please notify us in time!   We will try our best to revise this manuscript.

Thank you very much for your attention and kind suggestions.

Sincerely Yours,

Ms. Yiping Zhang

Reviewer 2 Report

In my opinion this manuscript is very interesting but has some problem that need correcting

The aim of this research were to evaluate the effects of different preceding crops on clubroot disease of Chinese cabbage and soil microorganisms, to provide a theoretical basis for the ecological control of clubroot scientifically. In this experiment, soybeans, potato onions, and wheat were used as the preceding crops comparing with the local preceding crop garlic, determining the growth of Chinese cabbage, disease occurrence, soil chemical properties and changes in microbial community structure by using quantitative real-time polymerase chain reaction (PCR), soil microbial high- throughput sequencing and other methods. The research undertaken is very important in the context of sustainable and ecological agriculture.

In my opinion this manuscript is very interesting but has some problem that need correcting. My specific comments are as follow:

Line 92. Units should be used in the SI system. What does Potassium sulfate ... (10kg · hm-2) mean, I think it should be (10kg · ha-2)

Line 93-96 The assumption of the experiment is described only in 2018 and in the following text also the year 2019 is given (in 143 line). The methods described are inconsistent with the results.

Line 105 It should be indicated when the soil samples were collected for analysis. Is it at the end of the experiment or at the beginning?

Line 180 How did you get 24 samples when there were 12 variants of experiment. Whether 12 samples each year or what…? The research methodology is not described in detail.

Line 146 Should be "was" not "is"

Line 148 Figure 8. there are no letters A and B under the graphs and there are letters in the description.

Author Response

Response to Reviewer 2 Comments

Dear Reviewer,

Here is our revised manuscript (Manuscript ID: microorganisms-1652351) for “Microorganisms”. Response to the comments:

Reviewer reports:

In my opinion this manuscript is very interesting but has some problem that need correcting.

The aim of this research were to evaluate the effects of different preceding crops on clubroot disease of Chinese cabbage and soil microorganisms, to provide a theoretical basis for the ecological control of clubroot scientifically. In this experiment, soybeans, potato onions, and wheat were used as the preceding crops comparing with the local preceding crop garlic, determining the growth of Chinese cabbage, disease occurrence, soil chemical properties and changes in microbial community structure by using quantitative real-time polymerase chain reaction (PCR), soil microbial high- throughput sequencing and other methods. The research undertaken is very important in the context of sustainable and ecological agriculture.

In my opinion this manuscript is very interesting but has some problem that need correcting. My specific comments are as follow:

Point 1: Line 92. Units should be used in the SI system. What does Potassium sulfate ... (10kg · hm-2) mean, I think it should be (10kg · ha-2)

Response 1: Thanks for your observant pointing. We have corrected the units under your suggestion.

Point 2: Line 93-96 The assumption of the experiment is described only in 2018 and in the following text also the year 2019 is given (in 143 line). The methods described are inconsistent with the results.

Response 2: Thank you very much for your kind advices. We added the description of the experiment in 2019 in line 90.

Point 3: Line 105 It should be indicated when the soil samples were collected for analysis. Is it at the end of the experiment or at the beginning?

Response 3: Thank you very much for your responsible comments. Soil was collected when the preceding crops were harvested, and we have also explained in the article.

Point 4: Line 180 How did you get 24 samples when there were 12 variants of experiment. Whether 12 samples each year or what…? The research methodology is not described in detail.

Response 4: We are deeply sorry for the trouble that our unclear description has caused you to understand. The 12 samples were subjected to Miseq sequencing for the purpose of bacterial 16SRNA V3-V4 regions and fungal ITS regions, respectively. So there were 24 samples in total. To prevent confusion, we have described the samples of used for the determination of bacteria and fungi separately in line 191.

Point 5: Line 146 Should be "was" not "is"

Response 5: We have modified the sentence to “In addition, the dry weight on the ground of Chinese cabbage in 2019 was generally lower than in 2018” in line 152.

Point 6: Line 148 Figure 8. there are no letters A and B under the graphs and there are letters in the description.

Response 6: We are sorry for our negligence, we guessed what you pointed is Figure 1, and we have added A and B in figure 1.

Finally, the manuscript was revised based on the comments of reviewers and editors. All changes in the manuscript use tracking marks. If there is any shortage, please notify us in time! We will try our best to revise this manuscript.

Thank you very much for your attention and kind suggestions.

Sincerely Yours,

Ms. Yiping Zhang

Round 2

Reviewer 1 Report

Lines 270, 282, 285, 295, 720 and 729, the letter "P" of the significance remains capitalized.

Line 275 will have to be checked, change "in bacterial" by fungal.